

# Arthropod IGF, relaxin and gonadulin, putative orthologs of *Drosophila* insulin-like peptides 6, 7 and 8, likely originated from an ancient gene triplication

Jan A. Veenstra

INCIA UMR 5287 CNRS, Université de Bordeaux, Pessac, France

## ABSTRACT

**Background**. Insects have several genes coding for insulin-like peptides and they have been particularly well studied in *Drosophila*. Some of these hormones function as growth hormones and are produced by the fat body and the brain. These act through a typical insulin receptor tyrosine kinase. Two other *Drosophila* insulin-like hormones are either known or suspected to act through a G-protein coupled receptor. Although insulin-related peptides are known from other insect species, *Drosophila* insulin-like peptide 8, one that uses a G-protein coupled receptor, has so far only been identified from *Drosophila* and other flies. However, its receptor is widespread within arthropods and hence it should have orthologs. Such putative orthologs were recently identified in decapods and have been called gonadulins.

**Methodology**. In an effort to identify gonadulins in other arthropods public genome assemblies and short-read archives from insects and other arthropods were explored for the presence of genes and transcripts coding insulin-like peptides and their putative receptors.

**Results**. Gonadulins were detected in a number of arthropods. In those species for which transcriptome data from the gonads is available insect gonadulin genes are expressed in the ovaries and at least in some species also in the testes. In some insects differences in gonadulin expression in the ovary between actively reproducing and non-reproducing females differs more than 100-fold. Putative orthologs of *Drosophila* ilp 6 were also identified. In several non-Dipteran insects these peptides have C-terminally extensions that are alternatively spliced. The predicted peptides have been called arthropod insulin-like growth factors. In cockroaches, termites and stick insects genes coding for the arthropod insulin-like growth factors, gonadulin and relaxin, a third insulin-like peptide, are encoded by genes that are next to one another suggesting that they are the result of a local gene triplication. Such a close chromosomal association was also found for the arthropod insulin-like growth factor and gonadulin genes in spiders. Phylogenetic tree analysis of the typical insulin receptor tyrosine kinases from insects, decapods and chelicerates shows that the insulin signaling pathway evolved differently in these three groups. The G-protein coupled receptors that are related to the *Drosophila* ilp 8 receptor similarly show significant differences between those groups.

**Conclusion**. A local gene triplication in an early ancestor likely yielded three genes coding gonadulin, arthropod insulin-like growth factor and relaxin. Orthologs of these

Corresponding author
Jan A. Veenstra,
jan-adrianus.veenstra@u-bordeaux.fr

genes are now commonly present in arthropods and almost certainly include the *Drosophila* insulin-like peptides 6, 7 and 8.

## INTRODUCTION

Insulin may well be the best studied and perhaps the best known hormone, due to its essential role in the regulation of glucose homeostasis and its effective and widespread use to treat diabetes. Insulin is related to a number of other hormones with different functions, such as insulin-like growth factors, relaxin and Ins3-5. One of the interesting aspects of these hormones is their use of two structurally very different receptors, receptor tyrosine kinases (RTKs) and leucine-rich repeat G-protein coupled receptors (LGRs). Thus, whereas insulin and insulin-like growth factors (IGFs) act through an RTK, relaxin and Ins3 use an LGR for signal transduction. An intriguing question remains as to how this switch was made from one type of receptor to another, or alternatively whether the ancestral insulin used perhaps both types of receptors and during evolution its descendants became specific ligands for only one of the two receptors.

Like other hormones and neuropeptides, insulin was already present in the ancestral bilaterian that gave rise to both protostomes and deuterostomes. The first indication that an insulin-like peptide (ilp) might exist in protostomes was the observation that insulin enhances cell differentiation in cultured *Drosophila* cells (*Seecof & Dewhurst, 1974*). The identification of one ilp in the silkworm *Bombyx mori* that can break diapause (*Nagasawa et al., 1984*; *Nagasawa et al., 1986*) and another one in neuroendocrine cells known to produce a growth hormone in the pond snail *Lymnaea stagnalis* (*Smit et al., 1988*) reinforced the hypothesis that insulins may function as growth hormones in protostomes. Since then a large variety of invertebrates has yielded a still larger variety of ilps (e.g., *Murphy & Hu, 2013*; *Mizoguchi & Okamoto, 2013*; *Nässel & Van den Broeck, 2016*; *Yu, Han & Liu, 2020*).

In insects the ilps of *Drosophila* and the silkworm *Bombyx mori* have been extensively studied and these hormones are best known in fruit fly due to the genetic power that can be employed in this species. There are eight ilp genes in *Drosophila melanogaster*, which are referred to as *Drosophila* ilps 1-8. *Drosophila* ilps 1, 2, 3 and 5 are co-expressed in a single cell type of neuroendocrine cells of the brain (*Brogiolo et al., 2001*; *Grönke et al., 2010*). Of these ilp 2 seems to be the most important and it also seems to be expressed exclusively or predominantly in these brain neuroendocrine cells. *Drosophila* ilps 3 and 5 are also expressed in other tissues, e.g., ilp 3 is expressed in midgut muscle of both larvae and adults where its expression stimulates midgut growth in response to feeding (*O'Brien et al., 2011*). *Drosophila* ilp 1 has been shown to be expressed in the brain neuroendocrine cells, but its expression is largely limited to stages when the animal does not feed, i.e., metamorphosis and diapause (*Liu et al., 2016*). The expression of dilp 4 seems limited to the embryonic stage, while ilp 6 is expressed predominantly if not exclusively by the fat body (*Slaidina et*

*al., 2009*; *Okamoto et al., 2009b*). All these ilps are believed to activate the single *Drosophila* insulin RTK, while *Drosophila* ilps 7 and 8 are either known (ilp 8) or suspected (ilp 7) to activate *Drosophila* LGRs 3 and 4 respectively (*Vallejo et al., 2015*; *Gontijo & Garelli, 2018*; *Veenstra, Rombauts & Grbic, 2012*). *Drosophila* ilp 7 is expressed by neurons in the abdominal neuromeres in a sex specific manner (*Miguel-Aliaga, Thor & Gould, 2008*; *Yang et al., 2008*; *Castellanos, Tang & Allan, 2013*), while ilp 8 is expressed by the imaginal disks as well as the ovary and testes as shown by flyatlas (*Gontijo & Garelli, 2018*; *Liao & Nässel, 2020*).

The primary amino acid sequences of the *Drosophila* ilps vary considerably and this is also the case in other arthropod species that have multiple genes coding insulin-related peptides. There is not only large sequence variability within a species, but also between species. Only when species are relatively closely related is it possible to reliably identify orthologous genes in different species. However, while in most insects the A- and B-chains have thus quite variable amino acid sequences, this not the case for orthologs of *Drosophila* ilp 7. The strong conservation of the primary amino acid sequence of these peptides allows for easy identification of its orthologs, not only in other insect species, but also in other protostomes like various mollusks and even in some deuterostomes (*Veenstra, Rombauts & Grbic, 2012*). The strongly conserved primary amino acid sequence of these peptides suggests that it interacts with another receptor than the other ilps, perhaps in addition to the RTK. As some ilps act through a G-protein coupled receptor (GPCR), it seemed a distinct possibility that *Drosophila* ilp 7 and their orthologs might also stimulate a GPCR. Interestingly, genes coding LGR4 and its orthologs are present in the same genomes as those that have genes coding orthologs of *Drosophila* ilp 7. This holds not only for insects, but also other arthropods, mollusks and even some basal deuterostomes. Every genome that has a *Drosophila* ilp 7 ortholog also has a LGR4 ortholog and *vice versa* (*Veenstra, Rombauts & Grbic, 2012*; *Veenstra, 2014*; *Veenstra, 2019*). Furthermore, LGR3 and LGR4 are holomologs of vertebrate LGRs that use ilps as ligands. This means that the ligands for the LGR4s must be the *Drosophila* ilp 7 orthologs. Since these peptides are so different from the typical insect neuroendocrine insulins, it made sense to give it a different name. Earlier work on *Drosophila* suggested that it might have a role similar to relaxin in vertebrates (*Yang et al., 2008*) and since LGR4 is an homolog of the relaxin receptor (*Veenstra, 2014*), has also been called relaxin, but it might be better to call it arthropod or invertebrate relaxin.

*Drosophila* ilp 8 is another ilp (for review see *Gontijo & Garelli, 2018*) that acts through a leucine-rich repeat GPCR, LGR3 (*Garelli et al., 2015*; *Vallejo et al., 2015*; *Colombani et al., 2015*). However, whereas *Drosophila* ilp 7 orthologs have well conserved primary amino acid sequences, this is not the case for *Drosophila* ilp 8. Indeed, if it were not for the common presence of LGR3 orthologs in insect and other arthropod genomes one might believe that this peptide hormone evolved within the higher flies and is absent from other insects. The imaginal disks in *Drosophila* produce and release ilp 8 as long as they develop and also when they get damaged. When it is no longer released this is used by the brain as a signal to initiate metamorphosis (*Garelli et al., 2012*; *Colombani, Andersen & Léopold, 2012*; *Jaszczak et al., 2016*). *Drosophila* ilp 8 is furthermore produced by the testes and

ovaries (*Liao & Nässel, 2020*) and since imaginal disks are only present in holometabolous insects, it is tempting to speculate that the gonads are the original site of expression of orthologs of this peptide. I had previously suggested that the crustacean androgenic insulin-like peptide that stimulates premature sexual maturation in male crustaceans and can induce sex reversal in females, might be an ortholog of *Drosophila* ilp 8 (*Veenstra, 2016b*). However, more recently a fourth type of ilp was identified in two decapod species, that seem to be structurally more similar to *Drosophila* ilp 8 than the androgenic insulin-like peptides (*Chandler et al., 2017*). It has now been shown that these peptides, which have been called gonadulins, are generally present in decapods and commonly expressed by the gonads (*Veenstra, 2020*). Since gonadulins might be orthologs of *Drosophila* ilp 8 (*Veenstra, 2020*), it seemed worthwhile to look for this hormone in other arthropods. Analysis of arthropod genome and transcriptome sequences revealed that such peptides are not limited to decapods but are also present in insects as well as chelicerates.

During this analysis interesting new details of the putative orthologs of *Drosophila* ilp 6 were also encountered as well as evidence suggesting that the putative orthologs of *Drosophila* ilps 6, 7 and 8 arose from an ancestral gene triplication.

## MATERIALS & METHODS

The sratoolkit (https://trace.ncbi.nlm.nih.gov/Traces/sra/sra.cgi?view=software) in combination with Trinity (*Grabherr et al., 2011*) was used in the search for transcripts encoding peptides that might be somewhat similar to insulin in insect gonad transcriptome short read archives (SRAs). The method consisted of using the tblastn_vdb command from the sratoolkit to recover individual reads from transcriptome SRAs that show possible sequence homology with insulin-like molecules. Since insulin-like peptides have highly variable sequences the command is run with the -seg no and -evalue 100 options. Reads that are identified are then collected using the vdb-dump command from the sratoolkit. The total number of reads recovered is much smaller than those typically present in an SRA and this allows one to use Trinity on a normal desktop computer to make a mini-transcriptome of those reads. This transcriptome is than searched using the BLAST+ program (https://blast.ncbi.nlm.nih.gov/Blast.cgi?PAGE_TYPE=BlastDocs&DOC_TYPE=Download) for possible insulin transcripts. This first round usually yields numerous false positives and perhaps a few partial transcripts that look interesting. These promising but partial transcripts are then used as query using the blastn_vdb command from the sratoolkit on the same SRAs and reads are collected anew and Trinity is used to make another transcriptome that is again queried for the presence of insulin-like transcripts. In order to obtain complete transcript the blastn_vdb search may need to be repeated several times. Alternatively genes coding such transcripts were identified in genome assemblies using the BLAST+ program and Artemis (*Rutherford et al., 2000*). Once such transcripts had been found, it was often possible to find orthologs from related species. For example, once the honeybee gonadulin was found, it was much easier to find it in other Hymenoptera. The same methods were used to identify relaxin and C-terminally extended ilps, which have much better conserved primary amino acid sequences and consequently are more

easily identified, as well as their putative receptors. Whenever possible all sequences were confirmed in both genome assemblies and in transcriptome SRAs. In many cases transcripts for the various ilps and receptors were already present in genbank, although they were not always correctly identified. All these sequences are listed in Spreadsheet S1.

Expression was estimated by counting how many RNAseq reads in each SRA contained coding sequence for each of the genes. In order to avoid untranslated sequences of the complete transcripts, that sometimes share homologous stretches with transcripts from other genes and can cause false positives, only the coding sequences were used as query in the blastn_vdb command from the sratoolkit. This yielded the blue numbers in Spreadsheet S2. In order to more easily compare the different SRAs these numbers were then expressed as per million spots in each particular SRA. These are the bold black numbers in Spreadsheet S2.

For the expression of alternative aIGF (arthropod insulin-like growth factor) splice forms reads for each splice variant were first separately identified. Unique identifiers in these two sets were determined to obtain the total number of reads for aIGF. Those identifiers that were present in the initial counts for both splice forms were counted separately and subtracted from the initial counts of the two splice variants to obtain the number of reads specific for each isoform.

The various SRAs that were used are listed in the supplementary pdf file and were downloaded from https://www.ncbi.nlm.nih.gov/sra/. The following genome assemblies were also analyzed: *Aedes aegypti* (*Matthews et al., 2018*), *Blattella germanica* (*Harrison et al., 2018*), *Bombyx mori* (*Kawamoto et al., 2019*), *Galleria melonella* (*Lange et al., 2018*), *Glossina morsitans* (*Attardo et al., 2019*), *Hermetia illucens* (*Zhan et al., 2020*), *Latrodectus hesperus* (https://www.ncbi.nlm.nih.gov/genome/?term=Latrodectus+hesperus), *Mesobuthus martensii* (*Cao et al., 2013*), *Oncopeltus fasciatus* (*Panfilio et al., 2019*), *Parasteatoda tepidariorum* (*Schwager et al., 2017*), *Pardosa pseudoannulata* (*Yu et al., 2019*), *Periplaneta americana* (*Li et al., 2018*), *Stegodyphus dumicola* (*Liu et al., 2019*), *Tetranychus urticae* (*Grbic et al., 2011*), *Timema cristinae* (*Riesch et al., 2017*), *Tribolium castaneum* (*Herndon et al., 2020*) and *Zootermopsis nevadensis* (*Terrapon et al., 2014*). All genomes were downloaded from https://www.ncbi.nlm.nih.gov/genome/.

## Phylogenetic and sequence similarity trees

For the phylogenetic tree of the insulin RTKs sequences were aligned with clustal omega (*Sievers et al., 2011*). Using Seaview (*Gouy, Guindon & Gascuel, 2010*) only well aligned sequences were retained and saved as a fasta file. Fasttree2 (*Price, Dehal & Arkin, 2010*) was then used to produce a phylogenetic tree using the ./FastTreeDbl command with the following options: -spr 4 -mlacc 2 -slownni. The phylogenetic GPCR tree was constructed in the same way using only the transmembrane regions.

The amino acid sequences of arthropod ilps are not well conserved and hence one can not make phylogenetic trees, as it is impossible to know which amino acid residues can be reliably aligned apart from the cysteines. Even the latter can cause problems when the spacing between is not the same in different peptides. To compare the different ilps an unbiased method is needed. For this clustal omega was used to align the complete precursor

sequences and even though visual inspection with Seaview reveals very poor alignments, the alignment was not changed but saved as a fasta file and Fasttree was used with the same parameters as above to construct trees. Although such trees are not phylogenetic trees they do allow for an unbiased comparison of the various sequences. The trees so produced have been called sequence similarity trees. Note that the branch probabilities of such trees give useful information as to how reliable the grouping of the various ilp precursors is.

### Prediction of prepropeptides processing

Signal peptides were predicted using signal P-5.0 (*Almagro Armenteros et al., 2019*). No attempts were made to predict convertase cleavage sites in the gonadulin and aIGF precursors. Both of these putative hormones are likely not made by neuroendocrine cells. This implies that these hormones may not be exposed to neuroendocrine convertases and hence rules that describe how these convertases might cleave would not be applicable.

## RESULTS

### Gonadulin-like peptides are present in many arthropods

Peptides that share the typical location of six cysteine residues with insulin but are insufficiently similar to known insect ilps to be easily recognized as such, were identified in a large number of arthropods. Their sequences differ not only from other ilps, but are also very variable between them. As a consequence they are difficult to find in genome assemblies, unless a sequence from a not too distantly related species is available. This explains why searches were most successful when done in ovary and/or testes transcriptomes. Putative gonadulin orthologs were identified not only in insects, but also in several chelicerates, notably spiders, a spider mite and scorpions. A list of gonadulin propeptides in representative species is given in Fig. 1 and additional sequences are provided in Spreadsheet S1. It is evident that although these peptides have been given the same name, their sequences diverge even more than the neuroendocrine arthropod insulins. When constructing a sequence similarity tree from the insect ilps the gonadulins are well separated from the relaxins and the other insect ilps. Interestingly even the insulins and aIGFs are reasonably well separated, except for the precursors from highly evolved flies (*Drosophila* and *Glossina*) and the head louse (Fig. 2). When this is repeated on sequences from a set of arthropod ilps, the gonadulins are once again well separated from the other insulin related peptides (Fig. 3, Fig. S1).

Publicly available transcriptomes were used to explore in which tissues they are expressed. By nature such data is imperfect, as these transcriptomes were not made to answer the question where gonadulin, other insulin-related peptides or their putative receptors are expressed and hence such data is limited. Some of the more salient examples are illustrated in Table 1. In honeybees the ovaries of virgin queens do not express significant amounts of gonadulin, but those of egg-laying queens produce it in large quantities. In a single queen bumblebee larva transcriptome the gonadulin reads are at least 15 times more numerous than those in the three transcriptomes each for male and worker larvae (Spreadsheet S2). In the tsetse fly *Glossina morsitans* the gene is strongly expressed in ovaries of non-pregnant females, i.e., those that mature an egg, and hardly at all in pregnant/lactating females when
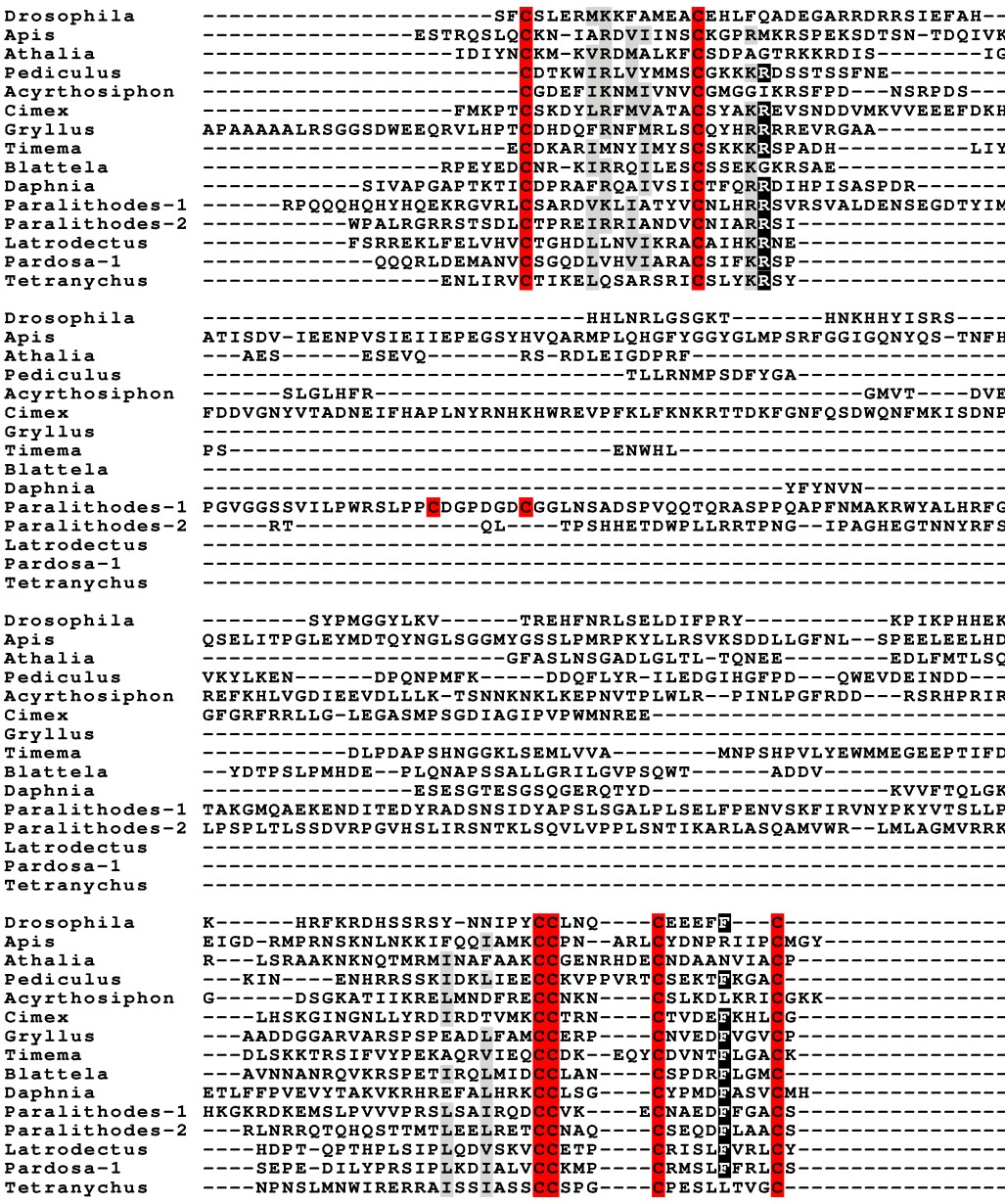

**Figure 1** **Gonadulin sequence alignment.** Note the very large sequence variability of the gonadulin propeptides. Cysteine residues are highlighted in red, other conserved amino residues in black and conserved substitutions in grey. Sequences are from Spreadsheet S1.

egg maturation is arrested. In the bugs *Rhodnius prolixus* and *Oncopeltus fasciatus* the ovaries also express this gene, as do ovaries and testes of the stick insect *Timema cristinae*, while short read archives of unfecundated eggs from *Blattella germanica* similarly contain large amounts of gonadulin reads. In the termite *Zootermopsis nevadensis*, gonadulin reads are abundant in reproducing females but rare or absent in alate females or reproducing males. The gene is also expressed in the testis of the American cockroach and possibly in the ovary

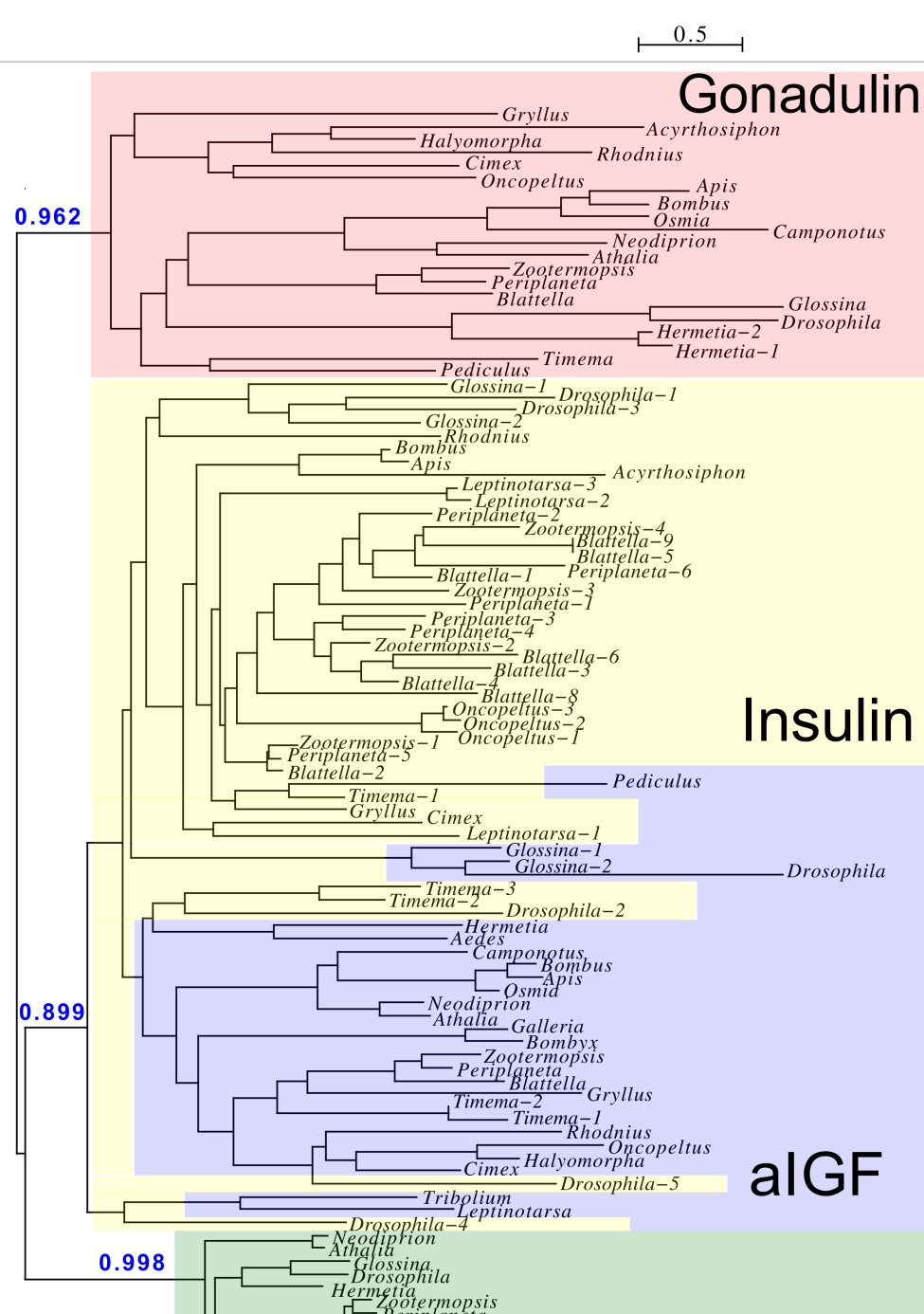

**Figure 2** **Insect ilp sequence similarity tree.** Note that the gonadulins and relaxins are well differentiated from the insulins and aIGFs, but that the latter are only partially separated from one another on the tree. Sequences are from Spreadsheet S1.

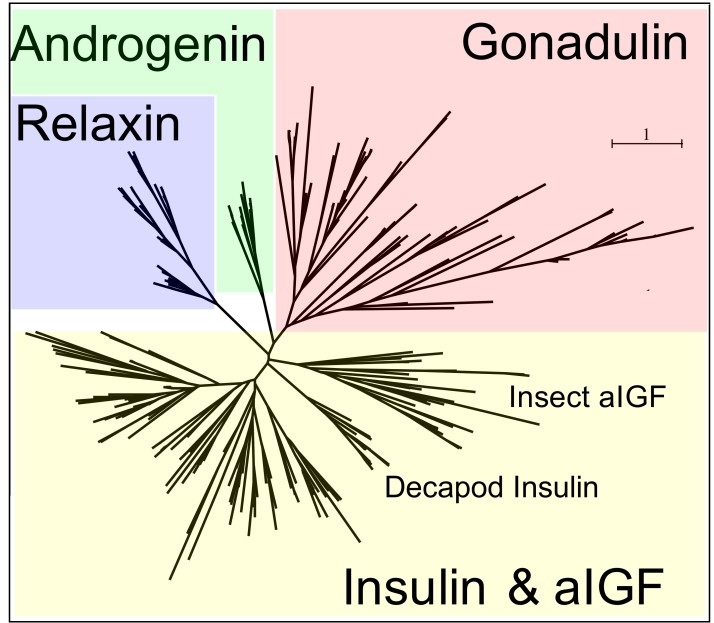

**Figure 3** **Radial arthropod ilp sequence similarity tree.** Note that the gonadulins cluster and are well separated from the other arthropod ilps. A more detailed sequence similarity tree with species names is present in the supplementary data as Fig. S1; sequences are from Spreadsheet S1.

as well, since it can be detected in whole body transcriptomes from females (Spreadsheet S2). However, as in decapods (*Veenstra, 2020*), in insects gonadulin expression is not limited to the gonads (Table 1; Spreadsheet S2). In the spider *Parasteatoda tepidariorum* ovary expression of gonadulin varied significantly between different samples (Table 1), and even larger variability in gonadulin expression has previously been reported for the crab *Portunus trituberculatus* (*Veenstra, 2020*). This shows that data from a single SRA are not necessarily informative as to the level of gonadulin expression in this organ. Interestingly, in some spider transcriptomes gonadulin expression is also observed in silk glands (Table 1, Spreadsheet S2).

### Arthropod insulin-like growth factors

Most insect ilps contain only a few amino acid residues after the sixth cysteine residue in the precursor and sometimes there are none, however some ilps have a long C-terminal extension. Such ilps are commonly present in hemimetabolous insects as well as several holometabolous species (Fig. 4, Fig. S2, Spreadsheet S1). In some species the C-terminal extension of these peptides are easily missed since if one ignores an intron donor site in the genome sequence the conceptual translation of such sequences predicts much smaller ilps that look similar to the well known *Drosophila* peptides. Nevertheless, analysis of RNAseq SRAs from several species shows that such intron donor sites are functional. These C-terminal extensions are coded by two additional exons that are not present in the typical insect neuroendocrine insulin genes. These extensions are not only commonly present, but also look similar to one another, providing further evidence that they are genuine

**Table 1  Gonadulin expression.**

| Species | Gonadulin | Tissue/organ |
|---|---|---|
| *Blattella germanica* | 0.00 | Male heads |
| | 7.30 | Female gonads and fat body |
| | 4.10 | Male gonads and fat body |
| | 493.70 | Non-fecundated eggs |
| *Timema cristinae* | | |
| | 0.00 | Head |
| | 23.30 | Testis |
| | 46.90 | Ovary |
| *Rhodnius prolixus* | | |
| | 78.40 | Ovary |
| | 1.40 | Testis |
| | 0.20 | CNS |
| | 3.00 | Antenna |
| *Apis mellifera* | | |
| | 0.20 | Ovary of virgin queen |
| | 61.60 | Ovary of normal egg-laying queen |
| | 96.70 | Ovary of normal egg-laying inhibted queen |
| | 59.00 | Ovary of normal egg-laying recovered queen |
| | 7.10 | Antenna |
| | 4.10 | Second thoracic ganglion |
| *Steatoda grossa* | | |
| | 0.00 | Cephalothorax |
| | 7.10 | Ovary with eggs |
| | 5.90 | Minor ampullate silk glands |
| | 3.30 | Tubuliform silk glands |
| *Parasteatoda tepidariorum* | | |
| | 2.00 | Ovary from SRR1824489 |
| | 0.00 | Ovary from SRR8755633 |
| | 0.00 | Ovary from SRR8755634 |

**Notes.**
Numbers are the number of gonadulin half reads per million in one or more transcriptome SRAs. This is a selection of the data from Spreadsheet S2.

parts of these ilps (Fig. 4). Furthermore, Trinity analysis of SRAs containing reads for such transcripts reveals that they are alternatively spliced which leads to the production of precursors with different C-terminal extensions. The difference often consists of the inclusion or exclusion of a sequence rich in dibasic amino acid residues, mostly arginines, that in many species has two characteristic cysteine residues. The alternative splice site is in the middle of what is usually the third coding exon, the last coding exon of these genes is less well conserved but contains a sequence that conforms more or less to the $GTVX_1PX_2(F/Y)$ consensus sequence. Such genes are present in species as diverse as cockroaches, termites, stick insects, beetles, bees, ants and moths (Fig. 5). Interestingly, in the stick insect *Timema cristinae* this gene underwent a local gene duplication, with

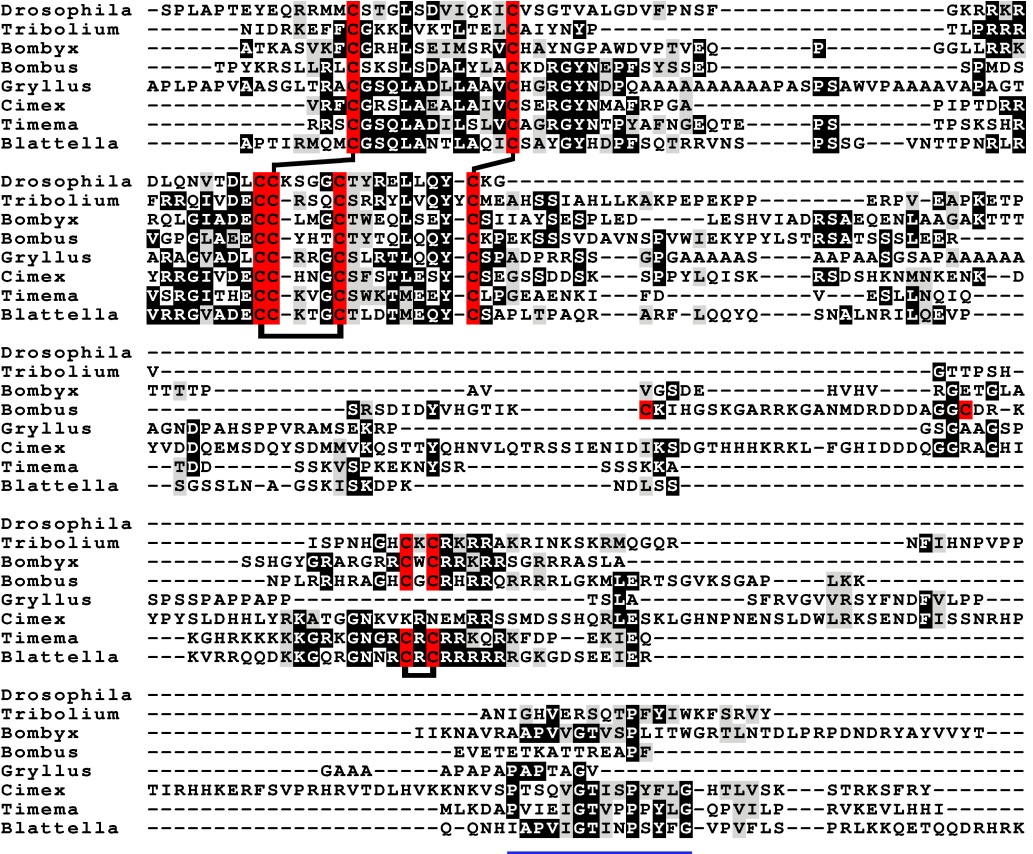

**Figure 4 aIGF sequence alignment.** Sequence alignment of some insect aIGF propeptides; where applicable only the isoforms with the arginine-rich sequences are shown. Note that these proteins have three different regions with sequence similarity: the insulin subsequence, the arginine-rich sequence containing an additional disulfide bridge and the C-terminal parts (underlined in blue), that is also somewhat conserved in *Gryllus rubens*, even though that species lacks an arginine-rich subsequence. Sequences are from Spreadsheet S1. Cysteine residues are indicated in red and the predicted disulfide bridges by lines. Other conserved amino acid residues are highlighted in black and conserved substitutions in grey.

one gene coding a peptide with the arginine-rich peptide sequence and the second one lacking it.

In hemiptera ilp genes exist that similarly code for C-terminally extended ilps that are alternatively spliced, but in those species the extended C-terminals of the predicted peptides are not as well conserved (Fig. S3). Other C-terminally extended ilps were found in spiders and scorpions, but in those species no evidence was found for alternative splicing. Such C-terminally extended ilps appear absent from decapods (*Veenstra, 2020*).

The term insulin-like growth factors was initially used as a description of substances in plasma that had insulin-like biological activity, but it is now mostly used as a name for the vertebrate hormones that are predominantly made in the liver. The use of the same term for both a group of molecules that have similar characteristics as well two specific hormones is confusing. This is particularly the case for insects, since hormones that have been called insulin-like growth factors are not necessarily orthologs of one another nor

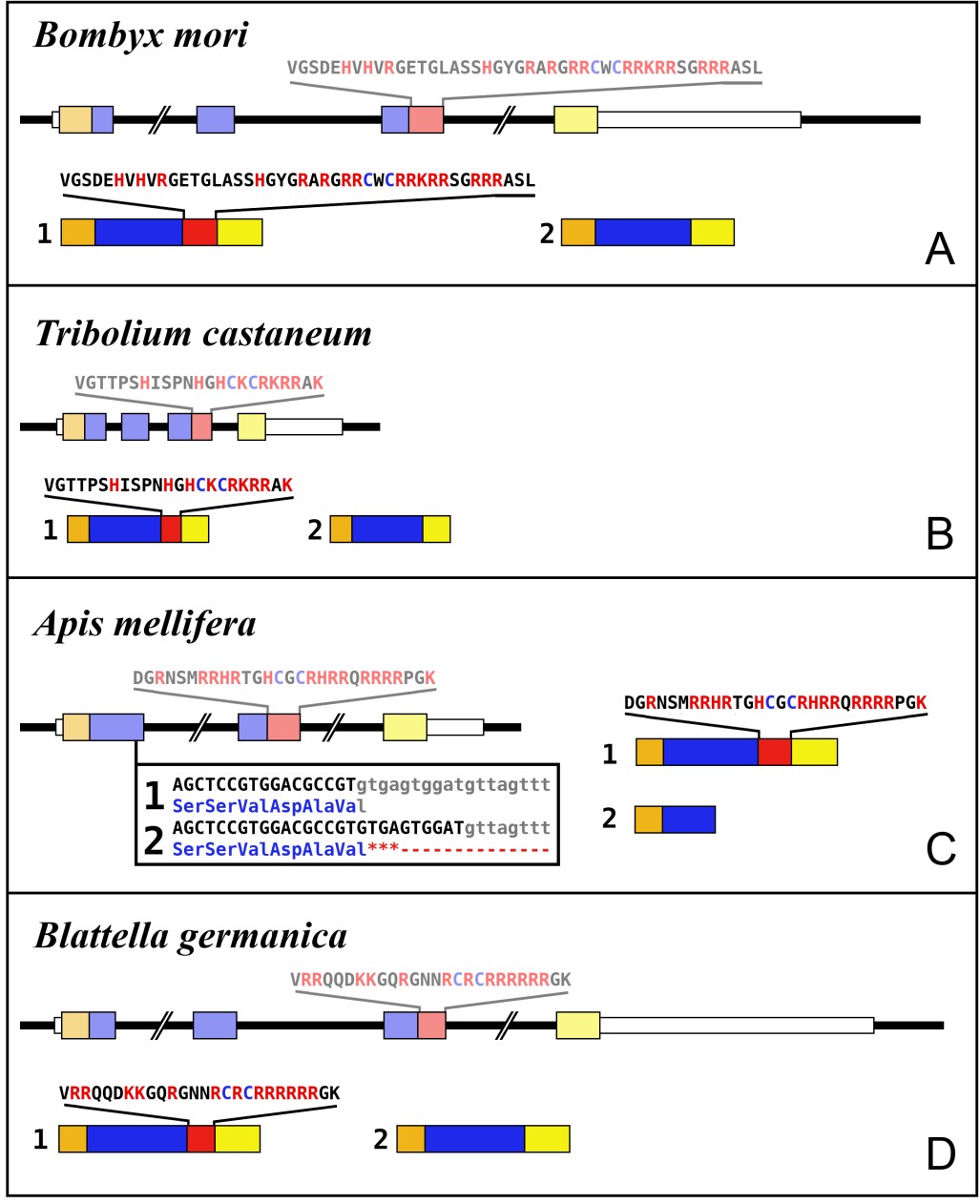

**Figure 5  Intron-exon structures of aIGF genes.** (A) *Bombyx mori*, (B) *Tribolium castaneum*, (C) *Apis mellifera*, (D) *Blattella germanica*. In the top part for each species the gene structure with the exon represented as small colored boxes and the introns as a black line. The orange brown color in the first exon corresponds to the coding sequence of the signal peptide and the blue corresponds to the insulin sequence, while the red color corresponds to the coding sequence of the arginine-rich region that is alternatively spliced in the two isoforms produced from these genes. The yellow exons contains coding sequence for the $GTVX_1PX_2(F/Y)$ consensus sequence. The amino acid sequence coded by this alternatively spliced DNA sequence is indicated. The numbers 1 and 2 show the coding sequences of the two mRNA species produced from these gene using the same colors as for the gene structures. Note that the structures of these genes are very similar, with the major differences being the size of the introns, some of which are very large, as indicated by interruption signs in the gene structures, and the loss of an intron in *Apis*. (continued on next page...)

**Figure 5 (…continued)**
The only other notable difference is that in the honey bee the second transcript is produced in a different fashion and only consists of one coding exon. The alternative splice site in this species have been indicated together with how this results in either splicing or the inclusion of a stop codon in the mRNA.

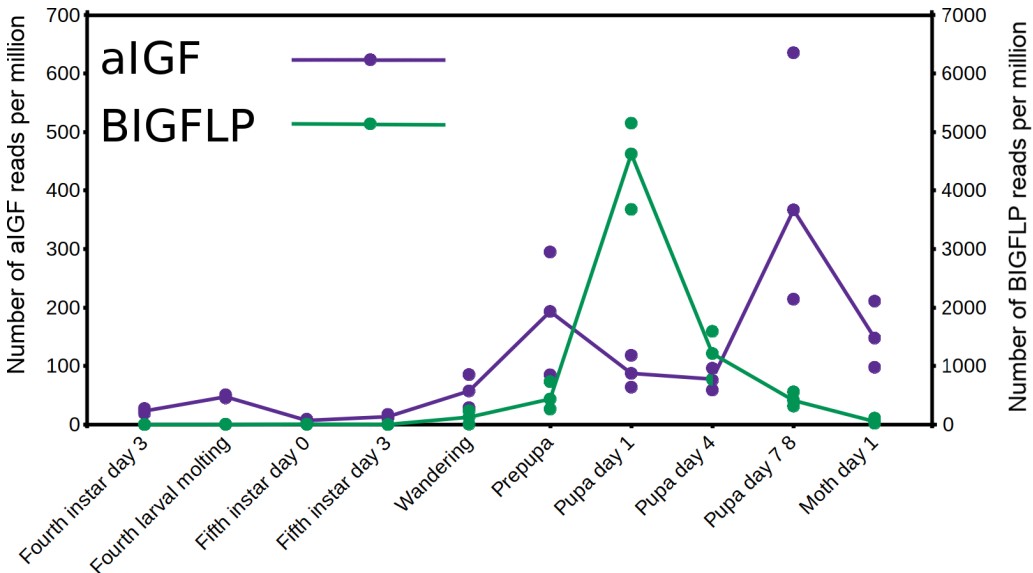

**Figure 6  Expression of aIGF and BIGFLP during development in *Bombyx mori*.**  Number of reads per million present in SRAs of a single series of experiments where there are three experiments for each developmental stage. The three values corresponds to the maximum, minimun and average of those three. Scale to the left is for aIGF, scale to the right for BIGFLP; note that the number of BIGFLP reads is much larger than those for aIGF.

of these vertebrate hormones. One of the two types of insect insulin-like growth factors, *Bombyx* IGF-like peptide (BIGFLP; *Okamoto et al., 2009a*), has only been found in *Bombyx mori*, although it can be expected to be present in other Lepidoptera as well. The insulin-like growth factor described above from several insect species seems to be commonly present in arthropods, including *Bombyx mori* and I propose to call it arthropod insulin-like growth factor (aIGF).

The data from an extended set of *Bombyx* transcriptome SRAs shows that in this species both insulin-like growth factors are expressed by the fat body, but the temporal patterns of expression of the two differ. Thus, aIGF is also expressed in larvae, when there is very little expression of BIGFLP while during the pupal stage their peaks of expression do not coincide (Fig. 6).

Alternative splicing of aIGF mRNA in the silkworm and the honey bee is variable between the different samples (Fig. 7, Fig. S4). In *Bombyx* SRAs from adults show a relatively higher expression of the longer isoform that has the additional sequence rich in dibasic amino acid residues (Fig. S5), while in the honeybee those are more common in SRAs from the bodies of larvae destined to become queens as well as in samples from the Nasonov glands from nurse bees and in the only available sample of queen heads (Fig. S4). In most species

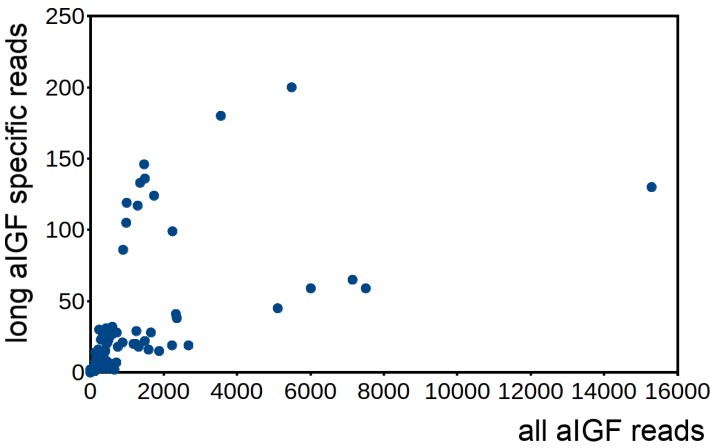

**Figure 7** **Differential expression of *Bombyx*. aIGF alternative splice forms.** Counts of reads specific for the long isoform of aIGF were plotted agains the total number of aIGF reads in all *Bombyx* SRAs (n = 253). Samples with high total aIGF counts show different levels of expression of the long isoform. Identification of some of the salient samples in this figure is provided in Fig. S5.

there are insufficient data or a single isoform may be predominantly expressed such as in the bumblebee (Spreadsheet S2).

## Is Drosophila ilp 6 an aIGF ortholog?

Given the absence of prominent sequence similarity between *Drosophila* ilp 6 and the aIGFs from more basal insect species, the obvious question is whether or not *Drosophila* ilp 6 is an ortholog of aIGF or whether it arose independently. The archetype aIGF gene contains four coding exons, the first two of which code for the insulin structure. The third consists of two parts, the first half which is present in both isoforms and the last part of this exon which is alternatively sliced. The fourth and last coding exon contains the $GTVX_1PX_2(F/Y)$ consensus sequence. During evolution, this basic pattern has been modified on several occasions. In *Oncopeltus* and other hemiptera the third coding exon was modified, while in both the mosquito *Aedes aegypti* and the honeybee, the intron between the first two coding exons was eliminated. In *Aedes* other changes occurred as well, but the gene can still be recognized as an aIGF gene by the presence of an exon that shows sequence similarity to the fourth coding exon of the aIGF genes. Soldier flies and robber flies have aIGF genes that are complete except for the third coding exons. It thus seems that the third coding exon was lost when Diptera evolved (Fig. 8). However, whereas the aIGF genes in those flies can still be recognized as such by the presence of what once was the fourth coding exon of a classical aIGF gene, this exon is not only absent from *Drosophila*, but is missing from all Erenomeura (Fig. 8). There are two possible explanations for this absence, either this exon was also lost, or the entire aIGF gene was lost and a novel insulin-like growth factor-like gene evolved in those flies, perhaps not unlike the origin of BIGFLP in *Bombyx*. The amino acid sequences of the aIGFs from the most highly developed non-Eremoneura flies are very similar to those from the least evolved Erenomeura flies (Fig. S6). So the aIGF gene persisted but it lost the last coding exon reminiscent of a classical aIGF gene and the

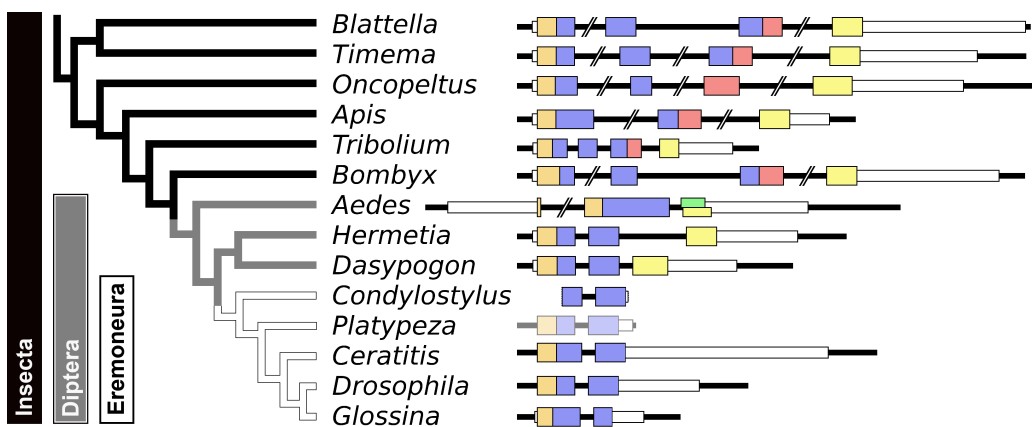

**Figure 8** **Evolution of the insect aIGF gene.** Schematic organization of insect aIGF genes in relation to their place on a simplified phylogenetic tree. The archetype of the insect aIGF gene consists of four coding exons. The first codes for the signal peptide (orange brown color) and the B-chain and part of the C-peptide, the second codes for the remainder of the C-peptide and the A-chain. The third coding exon codes for the first part of the C-terminal extension and has in the middle an alternative splice site allowing alternative splicing of the arginine rich sequene (in red). The last coding exon (yellow) codes for final part of the C-terminal extension that contains the $GTVX_1PX_2(F/Y)$ consensus sequence. Diptera have lost the third coding exon and the Eremoneura have also lost the fourth coding exon. Sequence comparison of the aILGF precursors from *Dasypogon, Hermetia, Condylostylus* and *Platypeza* are very similar, except for the loss of the last part (Fig. S6). The structures of the *Blattella, Apis, Tribolium* and *Bombyx* genes are from Fig. 5 and those of the *Oncopeltus* and *Aedes* genes from Fig. S3. The *Timema, Hermetia, Dasypogon, Ceratitis, Drosophila* and *Glossina* genes were produced in the same fashion. The *Condylostylus* gene could only partially constructed from genome and transcriptome SRAs. The *Platypeza* gene has been made translucent, as is only inferred; it is on a transcript (GCGU01007956.1) and assumes that its structure is identical to those of its closest relatives.

only physico-chemical characteristic that separates aIGF from the neuroendocrine insulins is the very short sequence connecting the putative A- and B-chains.

## Arthropod relaxins

The arthropod relaxins are known from a large number of arthropods, although they are lacking in many insect species. As noted previously they have by far the best conserved amino acid sequences in both the A- and B-chains of all the protostomian ilps and paralogs of this particular peptide are also found in other protostomians and even some basal deuterostomes. Neuropeptides containing cysteine residues typically have them in pairs, since as soon a cysteine containing peptide enters the endoplasmatic reticulum these residues get oxidized and form disulfide bridges. For this reason it is very surprising to see a decapod relaxin having seven cysteine residues (Fig. 9). The presence of this particular cysteine is unlikely to be an artifact as it is found in relaxins from a number of decapod species from different orders (Fig. S7). Although most arthropods appear to have a single relaxin gene, two scorpion genomes have two such genes; in both cases the predicted amino acid sequence of one of the relaxins seems less conserved (*Veenstra, 2016a*).

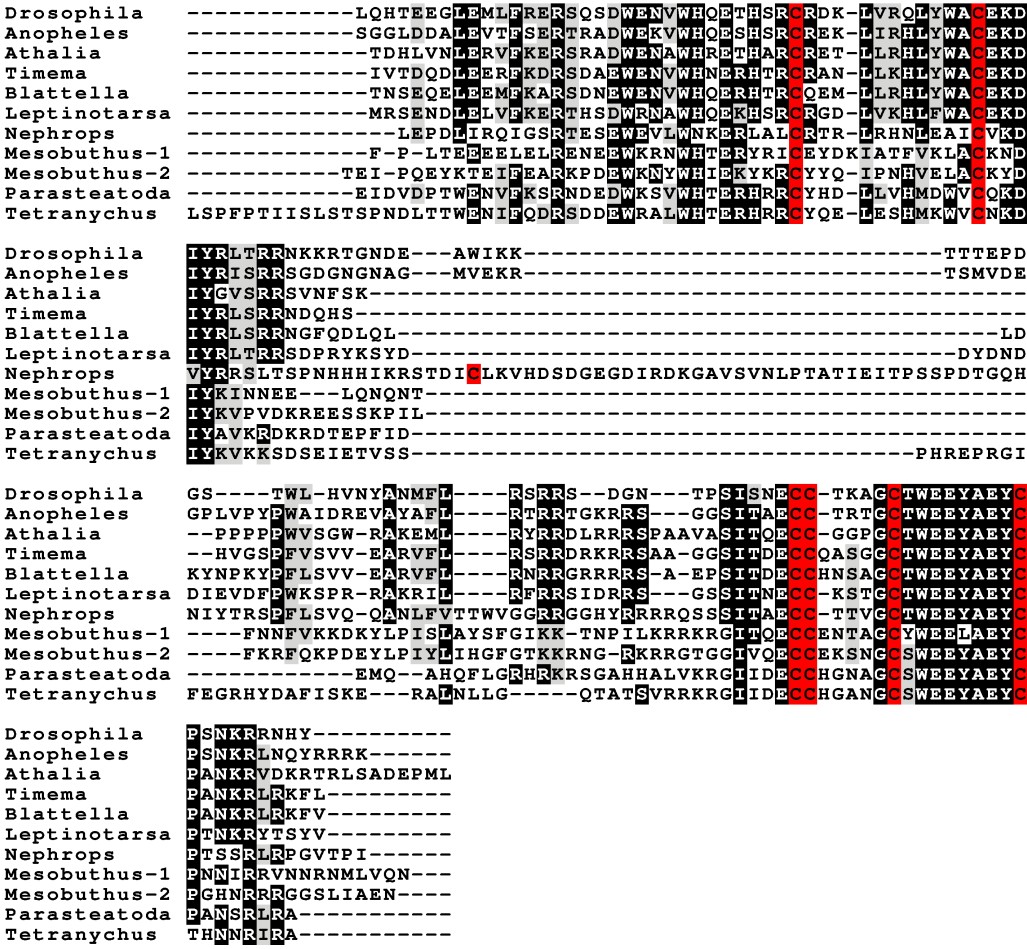

**Figure 9** **Arthropod relaxin sequence alignment.** Note how the relaxin propeptides are much better conserved than the either the gonadulins or the aIGFs. Also note that the exception is the single decapod relaxin that has also an additional cysteine residue. This is not a sequencing or other technical error, as the same residue is also found in several others decapods from different decapod orders (Fig. S7). Cysteine residues are highlighted in red, other conserved amino acid residues in black and conservative substitutions in grey. Sequences are provided in Spreadsheet S1 and from *Veenstra (2020)*.

## Gonadulin, relaxin and aIGF likely evolved from a local gene triplication

The very large primary amino acid sequence variability of arthropod ilps makes it difficult to establish their phylogenetic affinities. Sequence similarity trees show that the gonadulins resemble one another and hence may share a common ancestor, but such trees do not provide details. Synteny offers another means to establish evolutionary relationships and, as chance has it, the genomes of some species suggests that aIGF, gonadulin and arthropod relaxin originated from an ancient gene triplication. Thus the three genes coding gonadulin, aIGF and relaxin are located next to one another in the genomes of the German cockroach and the termite *Zootermopsis nevadensis*. In the stick insect *Timema cristinae* the aIGF underwent a local duplication (see above) and the four genes coding the two aIGFs,

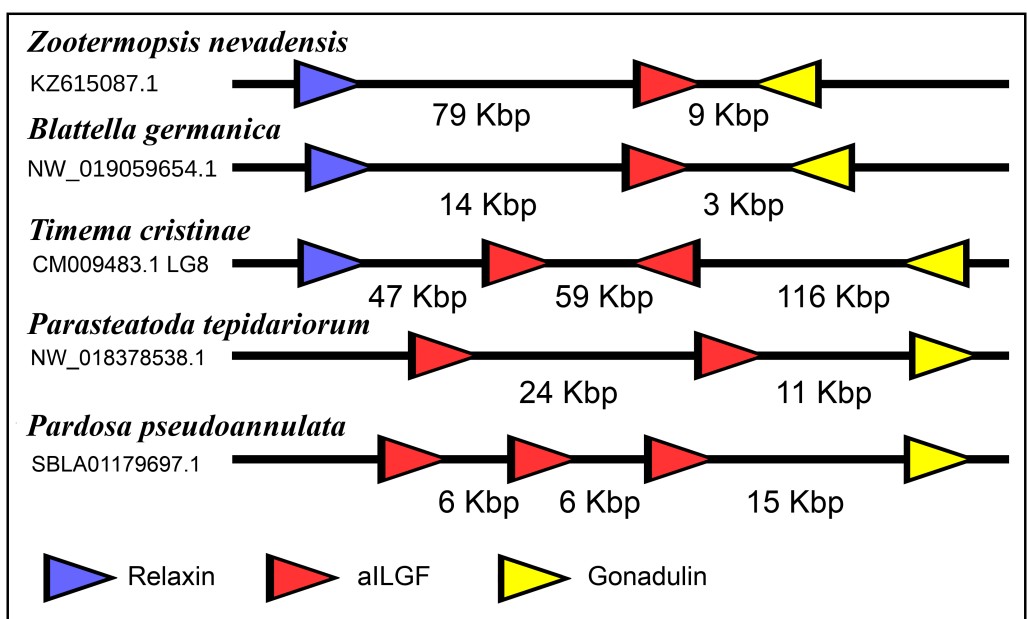

**Figure 10  Synteny of arthropod ilp genes.** Schematic representation of relaxin, gonadulin and aIGF genes in three insect and two spider species. Genbank accession numbers are indicated below the species name. Arrowheads indicate the direction of transcription. The numbers in Kbp indicate the distances between the coding regions of neighboring genes.

gonadulin and relaxin, are also next to one another. Furthermore, in the genomes from the spiders *Parasteatoda tepidariorum* and *Pardosa pseudoannulata* the gonadulin genes are next to an aIGF gene, which in these spiders has undergone one or more local duplications (Fig. 10). Such a close genomic association of these genes strongly suggests that they arose from a local gene triplication.

## Receptors

Two different types of receptors are used by ilps, LGRs and RTKs. Insects have two different types of insulin RTKs, that have been labeled A and B. Most species have one gene coding a type A and one gene coding for a B type B RTK. However, cockroaches, termites and at least some stick insects have two B type RTKs and the American cockroach has also two type A RTKs. The genome assembly of the American cockroach shows that the two type A RTKs originated from a local gene duplication, as did the two type B RTKs. In Lepidoptera and Diptera all insulin RTKs are type A and there is no type B RTK. However, Lepidoptera do have a second gene coding for a protein with clear structural similarity to insulin RTKs, but these proteins lack the tyrosine kinase domain. I have called those proteins Insulin RTK-like (IRTKL). This gene is more similar to the type B RTKs than type A and it may have evolved from those RTKs. The presence of an IRTKL coding gene in at least two genomes from Trichoptera, suggests that it may have evolved before the two groups diverged. Interestingly, the genome of the beetle *Tribolium castaneum* has genes for both a type A and a type B RTK, as well as for an IRTKL. No IRTKL orthologs was detected in any of other sequenced beetle genomes, but such an ortholog was found in

the transcriptome data of *Tenebrio molitor,* a species that is closely related to *Tribolium*. A phylogenetic tree of the various RTKs and IRTKL proteins suggests that the origin of the IRTKL proteins in Lepidoptera is ancient, but that of *Tribolium* is quite recent, indicating convergent evolution (Fig. 11). In chelicerates there are also two types of RTK, while there are four in decapods (*Veenstra, 2020*), however the various gene duplications of the arthropod RTKs occurred after these groups diverged (Fig. 11).

*Drosophila* LGR3 is activated by *Drosophila* ilp 8 and orthologs of this receptor are present in many but not all arthropods, always in a single copy. As described LGR4 is likely the arthropod relaxin receptor and is similarly present in many arthropods, usually in a single copy, but chelicerates have two copies and some spider have even three. LGR3 and LGR4 are closely related GPCRs together with another receptor that was first described from the pond snail *Lymnaea stagnalis* as GRL101 (*Tensen et al., 1994*) to stress its similarity with LGR3 and LGR4 it is called here LGR5. The strong sequence similarity of LGR5 with LGR3 and LGR4 suggests that it, like LGR3 and LGR4 might have an insulin-like ligand and for this reason it is included here. LGR5 is commonly present in arthropods. Unlike LGR3 and LGR4, that are missing in many insect species, LGR5 is consistently present in hemimetabolous insects, but it is completely absent from all holometabolous insects (the GRL101 from *Rhagoletis zephyria* [XP_017487580] appears to be a mite GPCR). In both chelicerates and decapods there are usually several paralogs (Fig. 12). All three of these receptors are widely expressed and it is difficult to discern clear expression patterns (Spreadsheet S2).

## DISCUSSION

I describe a number of novel arthropod ilps, the gonadulins and aIGFs, that are putative orthologs of *Drosophila* ilp 8 and ilp 6 respectively and I present evidence that the genes coding these peptides and relaxin are commonly present in arthropods and most likely originated from an ancient gene triplication.

There are four arguments that together suggests that the various gonadulins and *Drosophila* ilp 8 are indeed orthologs. First, the gonadulins cluster together with *Drosophila* ilp 8 on on a sequence similarity tree that bundles peptides with similar structures. It is clear that even though the amino acid sequences of the gonadulins are poorly conserved, they do resemble one another better than each one of them resembles any of the better known arthropod ilps. Secondly, for those members of this group where this could be determined, they are all made by the gonads, although this is not the only tissue expressing these peptides and expression by the gonads is variable. Thirdly, all species for which a gonadulin could be identified also have an ortholog of *Drosophila* LGR3, even though not all arthropod species have such a receptor. Unfortunately, due to the large structural variability of the gonadulins, it was not always possible to demonstrate the existence of a gonadulin gene in each species that has an ortholog of LGR3. Nevertheless, no gonadulins were found in species lacking such a receptor. Finally, peptides that were identified as putative gonadulins, but that are present in species as distantly related as spiders on one hand and stick insects and cockroaches on the other, are produced by genes for which orthology is independently confirmed by synteny with aIGF genes.

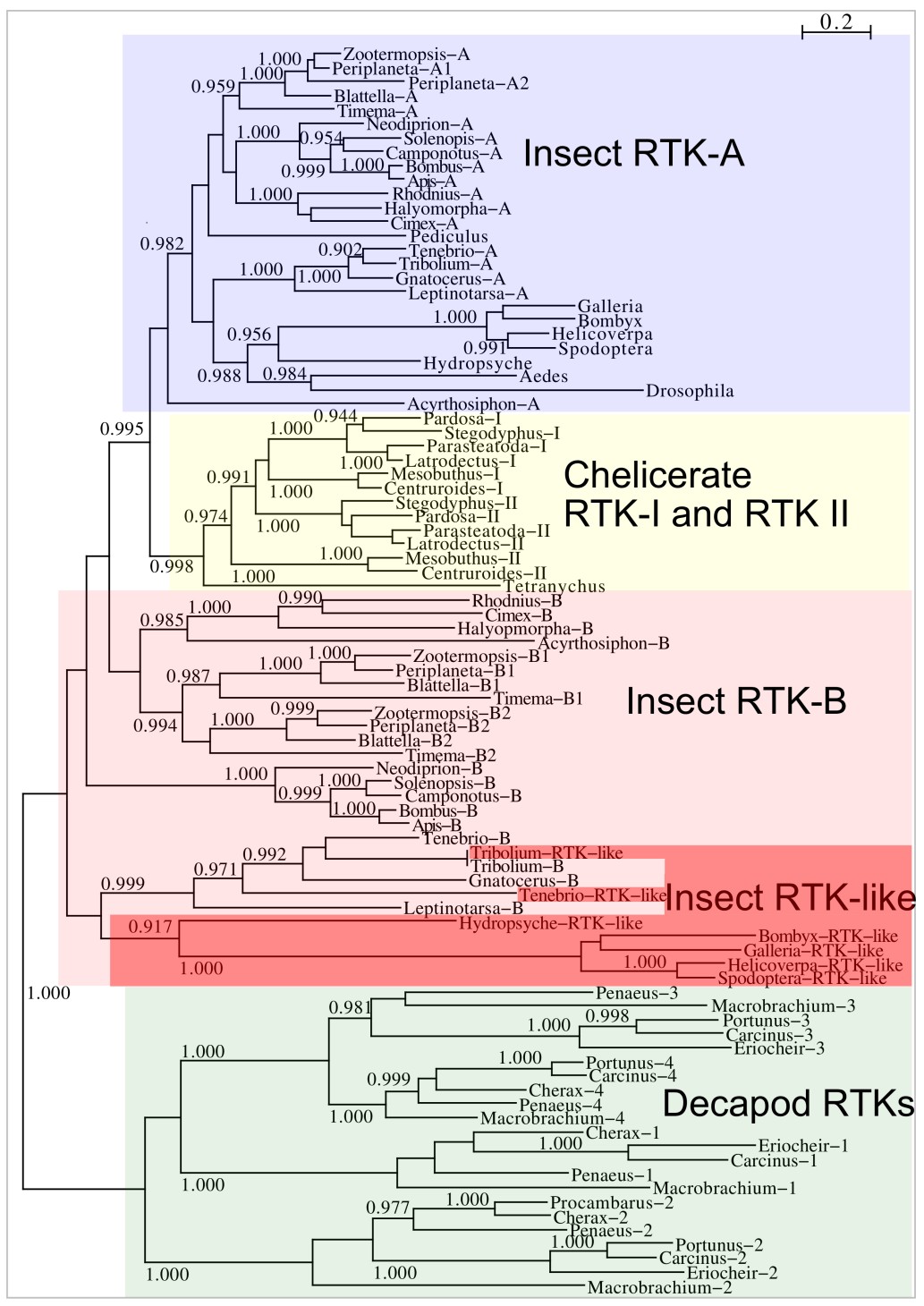

**Figure 11 Phylogenetic tree of arthropod insulin RTKs.** Phylogenetic tree of various arthropod RTKs. Note that the decapods, chelicerates and insect RTKs evolved independently. Only branch probabilities of more than 0.900 have been indicated. Sequences are from *Veenstra (2020)* and others provided in Spreadsheet S1.

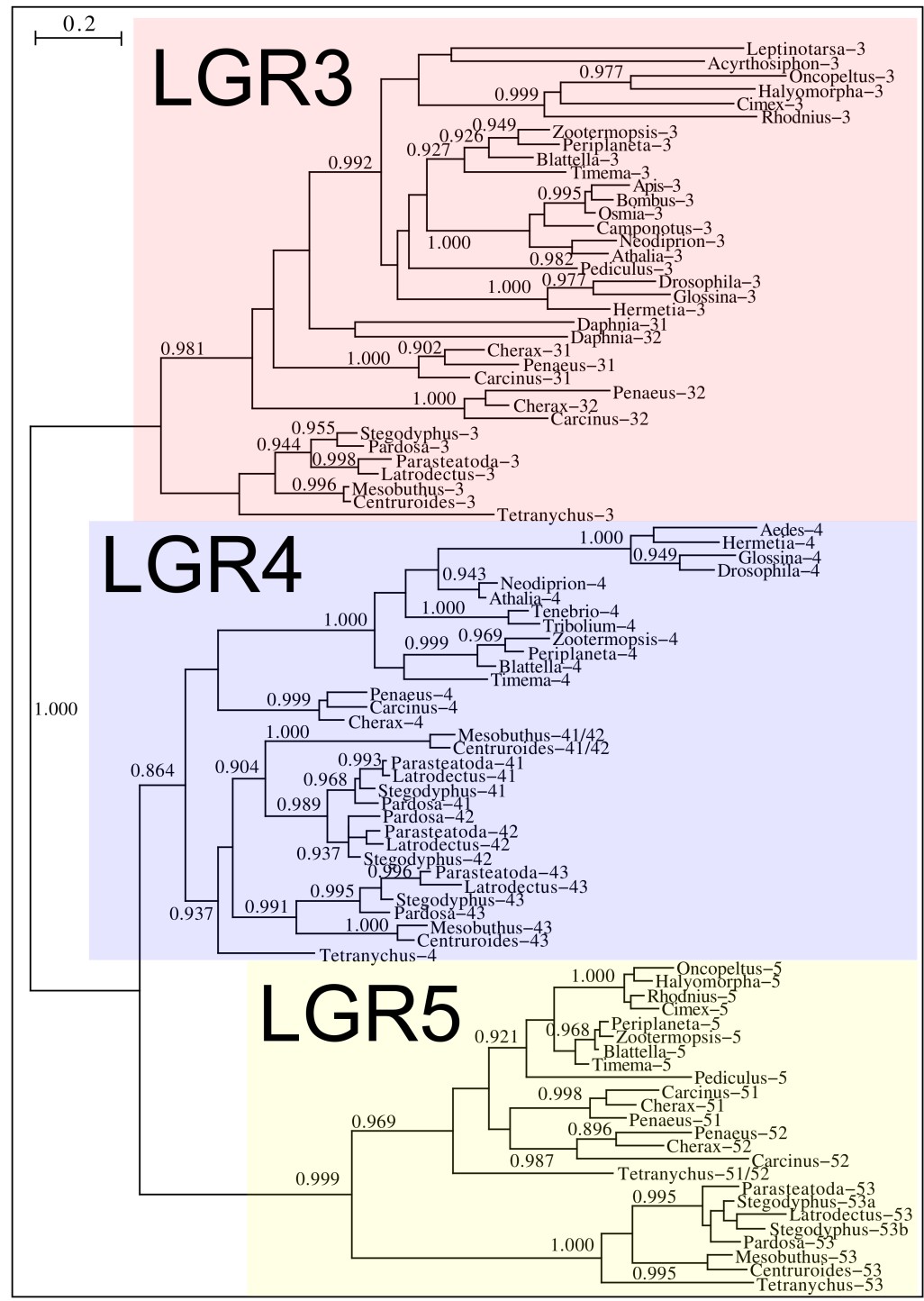

**Figure 12 Phylogenetic tree of putative arthropod insulin GPCRs.** Phylogenetic tree based exclusively on the transmembrane regions of various decapod LGRs that might function as receptors for insulin-related peptides. Only branch probabilities of more than 0.900 have been indicated. Sequences are from *Veenstra (2020)* and others provided in Spreadsheet S1.

The physiological significance of the presence of both aIGF and BIGFLP in *Bombyx* is an intriguing question, perhaps even more so as there seems to be only a single insulin receptor in this species that might induce an intracellular response. As shown by the data of an impressive set of transcriptome SRAs from this species, expression of the two differs during development (Fig. 6). Although the physiological meaning of this remains to be investigated, it is tempting to speculate that it is related to oogenesis and metamorphosis taking place simultaneously. In most insect species juvenile hormone and ecdysone play preponderant roles in the regulation of both processes, but using the same hormones for two different processes at the same time might be counterproductive and may have led to the evolution of BIGFLP.

It has been reported previously that, unlike *Drosophila*, most insect species have more than one insulin RTK (*Kremer, Korb & Bornberg-Bauer, 2016*). This is confirmed here and in the American cockroach there are actually four insulin RTKs. The presence of three insulin RTKs in termites has led to the suggestion that they might be involved in caste determination (*Kremer, Korb & Bornberg-Bauer, 2016*). Although this may be so, it is somewhat surprising in this context that the American cockroach, a close relative, has even four such receptors, yet has no caste system. In Lepidoptera and at least two tenebrionid beetles one of the putative insulin receptors lacks the tyrosine kinase domain, like the vertebrate IGF2 receptor and the *Drosophila* decoy insulin receptor (*Okamoto et al., 2013*). The *Drosophila* receptor also lacks the transmembrane domaine and is released into the hemolymph. These receptors may well have similars role in regulating the hemolymph ilp concentrations.

In some insects the expression of insulin RTKs have been studied sometimes together with the effects of inactivating one or both insulin RTKs by RNAi (*Wheeler, Buck & Evans, 2006*; *Sang et al., 2016*; *Okada et al., 2019*). Interestingly, in honeybee larvae that are changed from worker food to royal jelly, that is rich in proteins, it is the RTK A that is upregulated (*Wheeler, Buck & Evans, 2006*), whereas in two beetle species the effects of RTK inactivation by RNAi is much stronger for type A than for type B (*Sang et al., 2016*; *Okada et al., 2019*). In the honeybee and the beetle *Gnatocerus* these receptors have been indirectly linked to an aIGF peptide. In Diptera and in the head louse there is only a single RTK, which in both case are also type A and this is also the type that is most abundantly expressed in insects. This suggests that the A type insulin RTK is more important than the B type RTK.

The structural difference between neuroendocrine ilps and aIGFs, as illustrated by the large C-terminal extensions absent in the former but present in the latter, suggest the use of different receptors. Indeed, the similarities between insulin and vertebrate IGF on one hand and insect neuroendocrine ilps and aIGF on the other, are striking. Thus whereas the liver is the major tissue expressing IGF, in insects aIGF is expressed by the fat body, the tissue that performs the functions of both the vertebrate liver and adipose tissue. Furthermore, both IGF and aIGF are C-terminally extended insulin-like molecules and in both cases the primary transcripts produced are alternatively spliced (*Roberts Jr et al., 1987*). Finally, insulin and IGF act on two RTKs and many insects also seem to have two insulin RTKs, although structural similarity is insufficient proof that these are insulin

receptors, as illustrated by the insulin receptor–related receptor that functions as an alkali receptor (*Deyev et al., 2011*). Nevertheless, this suggests that the C-terminal extension of aIGF may allow for the differential activation of two different types of insulin RTKs and the presence of only one insulin receptor in Diptera may well explain why *Drosophila* ilp 6 has lost this C-terminal extension. Furthermore, it is perhaps no coincidence that the ilps that did not separate clearly into either an aIGF or an insulin branch on the sequence similarity tree (Fig. 2) are from *Drosophila*, *Glossina* and *Pediculus*. These are species that have only one insulin RTK and thus there would be no physiological need to maintain different molecular structures for peptides in order to preferentially activate one or the other of the two insulin RTKs.

The close genomic association of the gonadulin, relaxin and IGF genes in some arthropod genomes suggest that they originated from an ancient gene triplication. Although the primary amino acid sequences of the different gonadulins is limited, as discussed above, there is reason to believe they are also orthologs and the same holds for the various aIGFs. The *Drosophila* gonadulin (dilp 8) receptor has been shown to be *Drosophila* LGR3, while LGR4 must be an arthropod relaxin receptor. As demonstrated by the sequence similarity of their transmembrane regions, these LGRs and LGR5 are evolutionary cousins. Insect aIGFs on the other hand are known to stimulate RTKs. When neuropeptide genes undergo a local duplication the ligands they encode either keep using the same receptor or use paralogs that have their origin in a gene duplication of the receptor. The gene coding aIGF that uses an RTK is flanked on both sides by genes that code LGR ligands. The only reasonable explanation is that the original gene coding for an insulin-like peptide (the one that got triplicated) used both types of receptors, i.e. it had two receptors, both an LGR and an RTK.

It is clear that during evolution at least holometabolous species no longer have such a GPCR, as these species have at most only two such receptors, LGR3 and LGR4 (for gonadulin and relaxin respectively), while some species have none. Nevertheless, it is interesting to note that the hemimetabolous insects have LGR5, a receptor that is evolutionarily closely related to the receptors for gonadulin and relaxin. Thus LGR5 could be a second receptor for aIGF. If this were so, then the remarkable absence of LGR5 from holometabolous species might be related to the switch to holometaboly itself. Maggots and caterpillars undergo essentially linear growth, while in non-holometabolous species, growth is accompanied by development at the same time. Perhaps, LGR5 stimulated by aIGF is responsible for this.

Arthropod relaxin has an primary amino acid sequence that is much better conserved than that of the typical arthropod insulins or IGFs. As arthropod relaxin shares what seem to be the structural characteristics common to both aIGF and neuroendocrine insulins, it seems plausible that relaxin can also stimulate the insulin RTK. It is of interest in this respect that *Drosophila* relaxin, ilp7, binds to the decoy insulin receptor as well as other *Drosophila* ilps (*Okamoto et al., 2013*) and previous work suggested that it acts through the *Drosophila* RTK (*Linneweber et al., 2014*). When there is only a single receptor for a ligand, both can co-evolve and over time structures of both the ligand and its receptor may change. However, when the ligand activates two different receptors, all three components have to coevolve and one would expect that this would restrain the structures of all three

elements much more than when there is only a single receptor and such restraints would be the strongest on the ligand that activates both receptors. This might explain why arthropod relaxin is so well conserved.

It is tempting to speculate that orthologous genes have similar expression patterns and functions. In related species this is a reasonable hypothesis and the observed expression of aIGFs in cockroaches (*Castro-Arnau et al., 2019*) and beetles (*Okada et al., 2019*) is predominantly in the fat body and this seems to be the case for *Bombyx* aIGF too. What is likely the *Rhodnius* aIGF is also expressed by the fat body (*Defferrari, Orchard & Lange, 2015*) as is *Drosophila* ilp 6 (*Okamoto et al., 2009b*). This suggest that within insects these peptides have the same function. It is notable however, that at least in *Pardosa* spiders the fat body does not express aIGF but a specific insulin (*Yu, Han & Liu, 2020*), while aIGFs expression is limited to the cephalothorax. This suggests that in spiders aIGFs are expressed in the brain, as is indeed observed in the spider *Stegodyphus dumicola* (Spreadsheet S2). In decapods, which are more closely related to insects than chelicerates, no aIGFs were found (*Veenstra, 2020*). The phylogenetic tree of the various arthropod insulin RTKs also shows that the various paralogs of this receptor are not direct orthologs of one another, but must have evolved independently in each subphylum or even class. This within arthropods the functions of the various insulin-like peptides may be significantly different. It suggests that the apparent resemblance between insect neuroendocrine insulins and aIGF on one hand and insulin and IGF on the other could reflect a case of convergent evolution rather than one of orthology.

In the beetle *Gnatocerus cornutus* it has been shown that aIGF specifically stimulates the growth of a sexual ornament (*Okada et al., 2019*), while higher levels of aIGF are observed in honeybee larvae that are destined to become queens and thus develop functional ovaries (*Wheeler, Buck & Evans, 2006*). In *Gnatocerus* aIGF release depends on nutrition status and in honeybees protein rich royal jelly is associated with in increase of aIGF. Although we don't know as much detail for *Blattella* aIGF, its expression is strongly inhibited during starvation (*Castro-Arnau et al., 2019*). This suggests that in insects aIGF is released by the fat body in response to nutritious food.

The physiological function of gonadulin is less clear. Insulin and related peptides typically stimulate growth and reproduction, so its presence in the ovaries and testes suggests a function in reproduction. Its presence in unfecundated eggs of *Blattella* suggests that within the ovary it are the oocytes themselves that express gonadulin, likely the follicle cells that in *Drosophila* have been shown to express *Drosophila* ilp 8 (*Liao & Nässel, 2020*). In the crab *Portunus trituberculatus* gonadulin expression is on occasion very high in the gonads (*Veenstra, 2020*), but the very variable degree of expression makes it difficult to see this hormone as merely stimulating reproduction. The expression of gonadulin in hematopoetic tissue and the anterior proliferation center of the brain in *Procambarus clarkii* (*Veenstra, 2020*), neither suggest a role limited to reproduction but hints at a more general role in promoting growth. Spider silks are proteins and thus its production requires plenty of amino acids, not unlike vitellogenesis, or the development and reparation of imaginal disks. Gonadulin secreted by these organs might therefore suggest that, not unlike insulin, it stimulates growth, but more intensely and/or more localized. Such an intensified

stimulation of growth might be achieved by increasing not only the uptake of glucose as an energy substrate but also that of amino acids. Under this hypothesis, it might act as both an autocrine to stimulate uptake of metabolites and an endocrine to make these available and by doing so it might be able to stimulate growth of specific organs, such as imaginal disks and/or gonads that secrete it.

## CONCLUSIONS

A local gene triplication in an early ancestor likely yielded three genes coding gonadulin, arthropod insulin-like growth factor and relaxin. Orthologs of these genes are now commonly present in arthropods and almost certainly include the *Drosophila* insulin-like peptides 6, 7 and 8.

## ACKNOWLEDGEMENTS

I thank an anonymous reviewer and Jennifer Hackney Price for reviewing my manuscript, I also acknowledge the contributions made by all those who not only produced but also made available the numerous SRAs that I analyzed here. This work would neither have been possible without the various programs employed.

### Funding

This study was supported by institutional funding from the CNRS. The funders had no role in study design, data collection and analysis, decision to publish, or preparation of the manuscript.

### Grant Disclosures

The following grant information was disclosed by the author:
CNRS.

### Competing Interests

The authors declare there are no competing interests.

### Author Contributions

- Jan A Veenstra conceived and designed the experiments, performed the experiments, analyzed the data, prepared figures and/or tables, authored or reviewed drafts of the paper, and approved the final draft.

### Data Availability

The raw data are available in the Supplementary Files.

### Supplemental Information

Supplemental information for this article can be found online at http://dx.doi.org/10.7717/peerj.9534#supplemental-information.

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
