# Peer review of "Arthropod IGF, relaxin and gonadulin, putative orthologs of Drosophila insulin-like peptides 6, 7 and 8, likely originated from an ancient gene triplication"

_PeerJ, doi:10.7717/peerj.9534_

## Round 0.1 · original submission · Minor Revisions

Please take into consideration the reviewer's comments, and provide a revised manuscript and a detailed point-by-point rebuttal letter.

Reviewer 1 ·

Basic reporting

In this study, Veenstra has shown an intriguing insight into evolutionarily origin of insulin superfamily in arthropods. He showed several important arguments to suggest that three different groups of arthropod insulin family genes called gonadulin, arthropod insulin-like growth factor and relaxin likely originated from an ancestral gene triplication.

Experimental design

I have no quibbles with the quality and rigor of the data present and do not feel that there is a need for additional analysis.

Validity of the findings

This is an elegant and detailed study that addresses an evolutionarily aspects of the highly complicated and diverged insulin systems in arthropods.

Additional comments

1) Although the high conservation of the amino acid sequence in arthropod relaxin group, this is not the case for gonadulin group. If gonadulins indeed use Lgr3 as primary receptor, it is likely that arthropod gonadulins share high sequence similarity like arthropod relaxins. Does the author have any idea about this point? Is it possible that Lgr3 has another specific ligand other than gonadulin? or Is it possible that gonadulins have another receptor other than Lgr3?

2) I agree with the author’s opinion that aILGF and BIGFLP evolved independently. But it’s interesting that expressions of both Dilp6 and BIGFLP are regulated by ecdysteroid in the fat body (Okamoto et al., 2009a and 2009b; Slaidina et al., 2009). Does the author have any idea about this unusual similarity between Dilp6 and BIGFLP? Is it a convergent evolution? The independent evolution of BIGFLP and Dilp6 from different ancestral gene was also discussed in Okamoto et al., 2009a and Mizoguchi and Okamoto, 2013.

Other minor points as follows:

Line54: insulin-like growth factor → insulin-like growth factors (IGFs)

Line54: I don’t know why the author pick only Ins3 in here. Ins3 → insulin-like peptides (INSL) 3-5

Line56, 57, 426, 442, 444, 445, 447, 459, 482, 508: I recommend the author to use ‘IGFs’ instead of ‘ILGFs’ as an abbreviation of ‘insulin-like growth factors’.

Line89-90, 123-124: The author can cite Liao and Nassel, BioRxiv 2020 (https://doi.org/10.1101/2020.05.02.073585) instead of the Chintapalli et al., 2007 for the explanation of the expression and function of dilp8 in the ovary in Drosophila.

Line96: conversation → conservation

Line114: The author doesn’t have to cite Gontijo & Garelli, 2018 for two times in one sentence.

Line165: The ‘aILGF’ is firstly appear in here. The author needs to explain what is aILGF in the first appearance.

Line 288: I recommend the author to use ‘aIGF’ as an abbreviation of ‘arthropod insulin-like growth factor’.

Line295: The author needs to delete one of ‘sequence’.

·

Basic reporting

Table 1: As described in the introduction, many ILPs are associated with specific neuroendocrine cells in the CNS. Why are the reads shown for the antennae but not the brain or heads of honeybees? Showing reads from the brain or heads might be better suited to identify any CNS-associated transcripts.

Figures and tables: The meaning of the grey and black highlighted residues should be explained for all figures showing sequence alignments (Figures 1, 4, and 9). In figure 6, labeling the axes rather than describing them only in the figure legend would make interpretation a bit easier. Supplemental – spreadsheet 2: what is the significance of the yellow tabs?

There are a few instances in which the language could be improved for better clarity. Notably, what is meant by the ‘first additional exon’ (Line 267)? Further, it is unclear what is being referred to regarding the ‘insulin-like growth factor described above’ (Lines 283-288). Define aILGF the first time it is used (Line 165). In the methods section, aILGF seems to refer to alternatively-spliced ILGFs. However, it is later defined as arthropod ILGF (Lines 287-288). Several typos were also found (i.e. Lines 67, 355, 433, 439, 514 and the caption for Figure 11 – ‘decpods’).

Experimental design

This is an interesting and well-written paper that seems to focus on three goals: first, to identify insulin-like peptides in arthropods, second, to determine how widespread the two general classes of insulin receptors - receptor-tyrosine kinases and G protein-coupled receptors – are among arthropods, and third, to investigate the evolution of arthropod ILPs and their respective receptors. The study also clearly addresses the lack of knowledge of arthropod ILPs outside of typical model organisms such as Drosophila and Bombyx. Analysis, and interpretation seem reasonable based on data that were presented.

I have very limited experience with the bioinformatics methods utilized in this study, but I think clarity could be improved by adding a more thorough description of the analyses conducted using the SRA toolkit and Trinity, rather than directing readers to a separate paper which also does not seem to provide sufficient detail to repeat the analyses (lines 142-149). One thing that was unclear was whether Trinity could identify orthologs or was simply used to assemble reads. Was another tool (i.e. BLAST+ as described in Veenstra 2016) used to identify orthologs once the transcriptome was assembled?

Validity of the findings

Findings appear to be consistent with the data provided.

Additional comments

I appreciate how the author stepped through the main logical arguments that support the finding that dILP8 and the gonadulins are orthologs.

The identification of decapod relaxins containing an odd number of cysteine residues is intriguing (Lines 334-336 and Figure 9). Has this been observed for other neuropeptides? This observation is mentioned briefly in the results and the author should consider expanding upon it in the discussion section.

I also found the observation of a Bombyx-specific ILP interesting and wonder if there any other evidence that BIGFLP is found in other lepidoptera. Might the same types of analyses used in this ms be used to address this? Is there evidence of other species-specific ILPs? Understanding how common this phenomenon is would help to put the observation of the BIGFLP into perspective.

Reviewer 3 ·

Basic reporting

x

Experimental design

x

Validity of the findings

x

Additional comments

The author used bioinformatics approaches to analyze presence and evolution of various insulin-like peptides (ILPs) in different arthropods. This is an interesting paper which sounds more like a review. The interpretation of the results and hypotheses proposed in the paper are based on sequence similarities and available expression data from some species. These conclusions are rather speculative and need at least some experimental evidence which are beyond the scope of this paper. It is intriguing that “gonadulins” have only been found in hemimetabolan insects, but are missing in most Holometabola, except hymenopterans and flies. Similarly “relaxins” are restricted to representatives of a few non-related insect orders. Is it because these peptides and their receptors have been lost during evolution or (more likely) their sequences are very variable?
The author may consider mentioning obvious structural similarity between vertebrate IGF and some insect ILGF which are larger than regular insulins and composed of a single chain with three intramolecular disulfide bridges.

---

## Round 0.2 · accepted · Accept

The manuscript has improved in this review round and it is now accepted at PeerJ. Congratulations!